# Discriminative Metric Learning by Neighborhood Gerrymandering

**Shubhendu Trivedi,   David McAllester,   Gregory Shakhnarovich**
Toyota Technological Institute
Chicago, IL - 60637
{shubhendu,mcallester,greg}@ttic.edu

## Abstract

We formulate the problem of metric learning for $k$ nearest neighbor classification as a large margin structured prediction problem, with a latent variable representing the choice of neighbors and the task loss directly corresponding to classification error. We describe an efficient algorithm for exact loss augmented inference, and a fast gradient descent algorithm for learning in this model. The objective drives the metric to establish neighborhood boundaries that benefit the true class labels for the training points. Our approach, reminiscent of gerrymandering (redrawing of political boundaries to provide advantage to certain parties), is more direct in its handling of optimizing classification accuracy than those previously proposed. In experiments on a variety of data sets our method is shown to achieve excellent results compared to current state of the art in metric learning.

## 1   Introduction

Nearest neighbor classifiers are among the oldest and the most widely used tools in machine learning. Although nearest neighor rules are often successful, their performance tends to be limited by two factors: the computational cost of searching for nearest neighbors and the choice of the metric (distance measure) defining "nearest". The cost of searching for neighbors can be reduced with efficient indexing, e.g., [1, 4, 2] or learning compact representations, e.g., [13, 19, 16, 9]. We will not address this issue here. Here we focus on the choice of the metric. The metric is often taken to be Euclidean, Manhattan or $\chi^2$ distance. However, it is well known that in many cases these choices are suboptimal in that they do not exploit statistical regularities that can be leveraged from labeled data. This paper focuses on supervised metric learning. In particular, we present a method of learning a metric so as to optimize the accuracy of the resulting nearest neighbor classifier.

Existing works on metric learning formulate learning as an optimization task with various constraints driven by considerations of computational feasibility and reasonable, but often vaguely justified principles [23, 8, 7, 22, 21, 14, 11, 18]. A fundamental intuition is shared by most of the work in this area: an ideal distance for prediction is distance in the label space. Of course, that can not be measured, since prediction of a test example's label is what we want to use the similarities to begin with. Instead, one could learn a similarity measure with the goal for it to be a good proxy for the label similarity. Since the performance of $k$NN prediction often is the real motivation for similarity learning, the constraints typically involve "pulling" good neighbors (from the correct class for a given point) closer while "pushing" the bad neighbors farther away. The exact formulation of "good" and "bad" varies but is defined as a combination of proximity and agreement between labels. We give a formulation that facilitates a more direct attempt to optimize for the $k$NN accuracy as compared to previous work as far as we are aware. We discuss existing methods in more detail in section 2, where we also place our work in context.

In the $k$NN prediction problem, given a point and a chosen metric, there is an implicit hidden variable: the choice of $k$ "neighbors". The inference of the predicted label from these $k$ examples is trivial, by simple majority vote among the associated labels. Given a query point, there can possibly exist a very large number of choices of $k$ points that might correspond to zero loss: any set of $k$ points with the majority of correct class will do. We would like a metric to "prefer" one of these "good" example sets over any set of $k$ neighbors which would vote for a wrong class. Note that to win, it is not necessary for the right class to account for all the $k$ neighbors – it just needs to get more votes than any other class. As the number of classes and the value of $k$ grow, so does the space of available good (and bad) example sets.

These considerations motivate our approach to metric learning. It is akin to the common, albeit negatively viewed, practice of *gerrymandering* in drawing up borders of election districts so as to provide advantages to desired political parties, e.g., by concentrating voters from that party or by spreading voters of opposing parties. In our case, the "districts" are the cells in the Voronoi diagram defined by the Mahalanobis metric, the "parties" are the class labels voted for by the neighbors falling in each cell, and the "desired winner" is the true label of the training points associated with the cell. This intuition is why we refer to our method as *neighborhood gerrymandering* in the title.

Technically, we write $k$NN prediction as an inference problem with a structured latent variable being the choice of $k$ neighbors. Thus learning involves minimizing a sum of a structural latent hinge loss and a regularizer [3]. Computing structural latent hinge loss involves loss-adjusted inference — one must compute loss-adjusted values of both the output value (the label) and the latent items (the set of nearest neighbors). The loss augmented inference corresponds to a choice of worst $k$ neighbors in the sense that while having a high average similarity they also correspond to a high loss ("worst offending set of $k$ neighbors"). Given the inherent combinatorial considerations, the key to such a model is efficient inference and loss augmented inference. We give an efficient algorithm for *exact* inference. We also design an optimization algorithm based on stochastic gradient descent on the surrogate loss. Our approach achieves $k$NN accuracy higher than state of the art for most of the data sets we tested on, including some methods specialized for the relevant input domains.

Although the experiments reported here are restricted to learning a Mahalanobis distance in an explicit feature space, the formulation allows for nonlinear similarity measures, such as those defined by nonlinear kernels, provided computing the gradients of similarities with respect to metric parameters is feasible. Our formulation can also naturally handle a user-defined loss matrix on labels.

## 2 Related Work and Discussion

There is a large body of work on similarity learning done with the stated goal of improving $k$NN performance. In much of the recent work, the objective can be written as a combination of some sort of regularizer on the parameters of similarity, with loss reflecting the desired "purity" of the neighbors under learned similarity. Optimization then balances violation of these constraints with regularization. The main contrast between this body of work and our approach here is in the form of the loss.

A well known family of methods of this type is based on the Large Margin Nearest Neighbor (LMNN) algorithm [22] . In LMNN, the constraints for each training point involve a set of predefined "target neighbors" from correct class, and "impostors" from other classes. The set of target neighbors here plays a similar role to our "best correct set of $k$ neighbors" ($h^*$ in Section 4). However the set of target neighbors are chosen at the onset based on the euclidean distance (in absence of a priori knowledge). Moreover as the metric is optimized, the set of "target neighbors" is not dynamically updated. There is no reason to believe that the original choice of neighbors based on the euclidean distance is optimal while the metric is updated. Also $h^*$ represents the closest neighbors that have zero loss but they are not necessarily of the same class. In LMNN the target neighbors are forced to be of the same class. In doing so it does not fully leverage the power of the $k$NN objective. The role of imposters is somewhat similar to the role of the "worst offending set of $k$ neighbors" in our method ($\widehat{h}$ in Section 4). See Figure 2 for an illustration. Extensions of LMNN [21, 11] allow for non-linear metrics, but retain the same general flavor of constraints. There is another extension to LMNN that is more aligned to our work [20], in that they lift the constraint of having a static set of neighbors chosen based on the euclidean distance and instead learn the neighborhood.

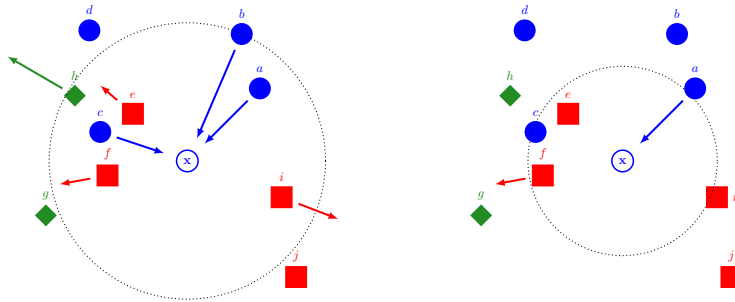

Figure 1: Illustration of objectives of LMNN (left) and our structured approach (right) for $k = 3$. The point $\mathbf{x}$ of class blue is the query point. In LMNN, the target points are the nearest neighbors of the same class, which are points $a, b$ and $c$ (the circle centered at $\mathbf{x}$ has radius equal to the farthest of the target points i.e. point b). The LMNN objective will push all the points of the wrong class that lie inside this circle out (points $e, f, h, i,$ and $j$), while pulling in the target points to enforce the margin. For our structured approach (right), the circle around $\mathbf{x}$ has radius equal to the distance of the farthest of the three nearest neighbors irrespective of class. Our objective only needs to ensure zero loss which is achieved by pushing in point $a$ of the correct class (blue) while pushing out the point having the incorrect class (point $f$). Note that two points of the incorrect class lie inside the circle ($e,$ and $f$), both being of class red. However $f$ is pushed out and not $e$ since it is farther from $\mathbf{x}$. Also see section 2.

The above family of methods may be contrasted with methods of the flavor as proposed in [23]. Here "good" neighbors are defined as all similarly labeled points and each class is mapped into a ball of a fixed radius, but no separation is enforced between the classes. The $k$NN objective does not require that similarly labeled points be clustered together and consequently such methods try to optimize a much harder objective for learning the metric.

In Neighborhood Component Analysis (NCA) [8], the piecewise-constant error of the $k$NN rule is replaced by a soft version. This leads to a non-convex objective that is optimized via gradient descent. This is similar to our method in the sense that it also attempts to directly optimize for the choice of the nearest neighbor at the price of losing convexity. This issue of non-convexity was partly remedied in [7], by optimization of a similar stochastic rule while attempting to collapse each class to one point. While this makes the optimization convex, collapsing classes to distinct points is unrealistic in practice. Another recent extension of NCA [18] generalizes the stochastic classification idea to $k$NN classification with $k > 1$.

In Metric Learning to Rank (MLR)[14], the constraints involve all the points: the goal is to push all the correct matches in front of all the incorrect ones. This again is not the same as requiring correct classification. In addition to global optimization constraints on the rankings (such as mean average precision for target class), the authors allow localized evaluation criteria such as Precision at $k$, which can be used as a surrogate for classification accuracy for binary classification, but is a poor surrogate for multi-way classification. Direct use of $k$NN accuracy in optimization objective is briefly mentioned in [14], but not pursued due to the difficulty in loss-augmented inference. This is because the interleaving technique of [10] that is used to perform inference with other losses based inherently on contingency tables, fails for the multiclass case (since the number of data interleavings could be exponential). We take a very different approach to loss augmented inference, using targeted inference and the classification loss matrix, and can easily extend it to arbitrary number of classes.

A similar approach is taking in [15], where the constraints are derived from triplets of points formed by a sample, correct and incorrect neighbors. Again, these are assumed to be set statically as an input to the algorithm, and the optimization focuses on the distance ordering (ranking) rather than accuracy of classification.

## 3    Problem setup

We are given $N$ training examples $X = \{\mathbf{x}_1, \ldots, \mathbf{x}_N\}$, represented by a "native" feature map, $\mathbf{x}_i \in \mathbb{R}^d$, and their class labels $\mathbf{y} = [y_1, \ldots, y_N]^T$, with $y_i \in [R]$, where $[R]$ stands for the set

$\{1, \ldots, R\}$. We are also given the loss matrix $\boldsymbol{\Lambda}$ with $\Lambda(r, r')$ being the loss incurred by predicting $r'$ when the correct class is $r$. We assume $\Lambda(r, r) = 0$, and $\forall (r, r')$, $\Lambda(r, r') \geq 0$.

We are interested in *Mahalanobis metrics*

$$D_{\mathbf{W}}(\mathbf{x}, \mathbf{x}_i) = (\mathbf{x} - \mathbf{x}_i)^T \mathbf{W} (\mathbf{x} - \mathbf{x}_i), \tag{1}$$

parameterized by positive semidefinite $d \times d$ matrices $\mathbf{W}$. Let $h \subset X$ be a set of examples in $X$. For a given $\mathbf{W}$ we define the distance score of $h$ w.r.t. a point $\mathbf{x}$ as

$$S_{\mathbf{W}}(\mathbf{x}, h) = - \sum_{\mathbf{x}_j \in h} D_{\mathbf{W}}(\mathbf{x}, \mathbf{x}_j) \tag{2}$$

Hence, the set of $k$ nearest neighbors of $\mathbf{x}$ in $X$ is

$$h_{\mathbf{W}}(\mathbf{x}) = \operatorname*{argmax}_{|h|=k} S_{\mathbf{W}}(\mathbf{x}, h). \tag{3}$$

For the remainder we will assume that $k$ is known and fixed. From any set $h$ of $k$ examples from $X$, we can predict the label of $\mathbf{x}$ by (simple) majority vote:

$$\widehat{y}(h) = \operatorname{majority}\{y_j : \mathbf{x}_j \in h\},$$

with ties resolved by a heuristic, e.g., according to 1NN vote. In particular, the $k$NN classifier predicts $\widehat{y}(h_{\mathbf{W}}(\mathbf{x}))$. Due to this deterministic dependence between $\widehat{y}$ and $h$, we can define the classification loss incured by a voting classifier when using the set $h$ as

$$\Delta(y, h) = \Lambda(y, \widehat{y}(h)). \tag{4}$$

## 4 Learning and inference

One might want to learn $\mathbf{W}$ to minimize training loss $\sum_i \Delta(y_i, h_{\mathbf{W}}(\mathbf{x}_i))$. However, this fails due to the intractable nature of classification loss $\Delta$. We will follow the usual remedy: define a tractable surrogate loss.

Here we note that in our formulation, the output of the prediction is a structured object $h_{\mathbf{W}}$, for which we eventually report the deterministically computed $\widehat{y}$. Structured prediction problems usually involve loss which is a generalization of the hinge loss; intuitively, it penalizes the gap between score of the correct structured output and the score of the "worst offending" incorrect output (the one with the highest score *and* highest $\Delta$).

However, in our case there is no single correct output $h$, since in general many choices of $h$ would lead to correct $\widehat{y}$ and zero classification loss: any $h$ in which the majority votes for the right class. Ideally, we want $S_{\mathbf{W}}$ to prefer *at least one* of these correct $h$s over all incorrect $h$s. This intuition leads to the following surrogate loss definition:

$$L(\mathbf{x}, y, \mathbf{W}) = \max_h [S_{\mathbf{W}}(\mathbf{x}, h) + \Delta(y, h)] \tag{5}$$

$$- \max_{h:\Delta(y,h)=0} S_{\mathbf{W}}(\mathbf{x}, h). \tag{6}$$

This is a bit different in spirit from the notion of margin sometimes encountered in ranking problems where we want all the correct answers to be placed ahead of all the wrong ones. Here, we only care to put *one* correct answer on top; it does not matter which one, hence the $\max$ in (6).

## 5 Structured Formulation

Although we have motivated this choice of $L$ by intuitive arguments, it turns out that our problem is an instance of a familiar type of problems: latent structured prediction [24], and thus our choice of loss can be shown to form an upper bound on the empirical task loss $\Delta$.

First, we note that the score $S_{\mathbf{W}}$ can be written as

$$S_{\mathbf{W}}(\mathbf{x}, h) = \left\langle \mathbf{W}, - \sum_{\mathbf{x}_j \in h} (\mathbf{x} - \mathbf{x}_j)(\mathbf{x} - \mathbf{x}_j)^T \right\rangle, \tag{7}$$

where $\langle \cdot, \cdot \rangle$ stands for the Frobenius inner product. Defining the *feature map*

$$\boldsymbol{\Psi}(\mathbf{x}, h) \triangleq - \sum_{\mathbf{x}_j \in h} (\mathbf{x} - \mathbf{x}_j)(\mathbf{x} - \mathbf{x}_j)^T, \tag{8}$$

we get a more compact expression $\langle \mathbf{W}, \boldsymbol{\Psi}(\mathbf{x}, h) \rangle$ for (7).

Furthermore, we can encode the deterministic dependence between $y$ and $h$ by a "compatibility" function $A(y, h) = 0$ if $y = \widehat{y}(h)$ and $A(y, h) = -\infty$ otherwise. This allows us to write the joint inference of $y$ and (hidden) $h$ performed by $k$NN classifier as

$$\widehat{y}_{\mathbf{W}}(\mathbf{x}), \widehat{h}_{\mathbf{W}}(\mathbf{x}) = \operatorname*{argmax}_{h, y} \left[ A(y, h) + \langle \mathbf{W}, \boldsymbol{\Psi}(\mathbf{x}, h) \rangle \right]. \tag{9}$$

This is the familiar form of inference in a latent structured model [24, 6] with latent variable $h$. So, despite our model's somewhat unusual property that the latent $h$ completely determines the inferred $y$, we can show the equivalence to the "normal" latent structured prediction.

## 5.1 Learning by gradient descent

We define the objective in learning $\mathbf{W}$ as

$$\min_{\mathbf{W}} \|\mathbf{W}\|_F^2 + C \sum_i L(\mathbf{x}_i, y_i, \mathbf{W}), \tag{10}$$

where $\| \cdot \|_F^2$ stands for Frobenius norm of a matrix.[1] The regularizer is convex, but as in other latent structured models, the loss $L$ is non-convex due to the subtraction of the max in (6). To optimize (10), one can use the convex-concave procedure (CCCP) [25] which has been proposed specifically for latent SVM learning [24]. However, CCCP tends to be slow on large problems. Furthermore, its use is complicated here due to the requirement that $\mathbf{W}$ be positive semidefinite (PSD). This means that the inner loop of CCCP includes solving a semidefinite program, making the algorithm slower still. Instead, we opt for a simpler choice, often faster in practice: stochastic gradient descent (SGD), described in Algorithm 1.

---

**Algorithm 1:** Stochastic gradient descent

**Input**: labeled data set $(X, Y)$, regularization parameter $C$, learning rate $\eta(\cdot)$
initialize $\mathbf{W}^{(0)} = \mathbf{0}$
**for** $t = 0, \ldots,$ *while not converged* **do**
    sample $i \sim [N]$
    $\widehat{h}_i = \operatorname{argmax}_h \left[ S_{\mathbf{W}^{(t)}}(\mathbf{x}_i, h) + \Delta(y_i, h) \right]$
    $h_i^* = \operatorname{argmax}_{h:\Delta(y_i, h)=0} S_{\mathbf{W}^{(t)}}(\mathbf{x}_i, h)$
    $\delta \mathbf{W} = \left[ \dfrac{\partial S_{\mathbf{W}}(\mathbf{x}_i, \widehat{h}_i)}{\partial \mathbf{W}} - \dfrac{\partial S_{\mathbf{W}}(\mathbf{x}_i, h_i^*)}{\partial \mathbf{W}} \right] \Big|_{\mathbf{W}^{(t)}}$
    $\mathbf{W}^{(t+1)} = (1 - \eta(t)) \mathbf{W}^{(t)} - C\delta \mathbf{W}$
    project $\mathbf{W}^{(t+1)}$ to PSD cone

---

The SGD algorithm requires solving two inference problems ($\widehat{h}$ and $h^*$), and computing the gradient of $S_{\mathbf{W}}$ which we address below.[2]

### 5.1.1 Targeted inference of $h_i^*$

Here we are concerned with finding the highest-scoring $h$ constrained to be compatible with a given target class $y$. We give an $O(N \log N)$ algorithm in Algorithm 2. Proof of its correctness and complexity analysis is in Appendix.

**Algorithm 2:** Targeted inference

---

**Input**: $\mathbf{x}$, $\mathbf{W}$, target class $y$, $\tau \triangleq [\![\text{ties forbidden}]\!]$
**Output**: $\operatorname{argmax}_{h:\widehat{y}(h)=y} S_{\mathbf{W}}(\mathbf{x})$
Let $n^* = \lceil \frac{k+\tau(R-1)}{R} \rceil$      // min. required number of neighbors from y
$h := \varnothing$
**for** $j = 1, \ldots, n^*$ **do**
    $h := h \cap \displaystyle\operatorname*{argmin}_{\mathbf{x}_i : y_i = y, i \notin h} D_{\mathbf{W}}(\mathbf{x}, \mathbf{x}_i)$
**for** $l = n^* + 1, \ldots, k$ **do**
    **define** $\#(r) \triangleq |\{i : \mathbf{x}_i \in h, y_i = r\}|$   // count selected neighbors from class r
    $h := h \cap \displaystyle\operatorname*{argmin}_{\mathbf{x}_i : y_i = y, \text{ or } \#(y_i) < \#(y) - \tau, i \notin h} D_{\mathbf{W}}(\mathbf{x}, \mathbf{x}_i)$
**return** $h$

---

The intuition behind Algorithm 2 is as follows. For a given combination of $R$ (number of classes) and $k$ (number of neighbors), the minimum number of neighbors from the target class $y$ required to allow (although not guarantee) zero loss, is $n^*$ (see Proposition 1 in the App. The algorithm first includes $n^*$ highest scoring neighbors from the target class. The remaining $k - n^*$ neighbors are picked by a greedy procedure that selects the highest scoring neighbors (which might or might not be from the target class) while making sure that no non-target class ends up in a majority.

When using Alg. 2 to find an element in $H^*$, we forbid ties, i.e. set $\tau = 1$.

### 5.1.2   Loss augmented inference $\widehat{h}_i$

Calculating the $\max$ term in (5) is known as loss augmented inference. We note that

$$\max_{h'} \langle \mathbf{W}, \mathbf{\Psi}(\mathbf{x}, \mathbf{h}') \rangle + \Delta(y, h') = \max_{y'} \Big\{ \underbrace{\max_{h' \in H^*(y')} \langle \mathbf{W}, \mathbf{\Psi}(\mathbf{x}, \mathbf{h}') \rangle}_{= \langle \mathbf{W}, \mathbf{\Psi}(\mathbf{x}, \mathbf{h}^*(\mathbf{x}, \mathbf{y}')) \rangle} + \Lambda(y, y') \Big\} \quad (11)$$

which immediately leads to Algorithm 3, relying on Algorithm 2. The intuition: perform targeted inference for each class (as if that were the target class), and the choose the set of neighbors for the class for which the loss-augmented score is the highest. In this case, in each call to Alg. 2 we set $\tau = 0$, i.e., we allow ties, to make sure the $\operatorname{argmax}$ is over all possible $h$'s.

---

**Algorithm 3:** Loss augmented inference

---

**Input**: $\mathbf{x}$, $\mathbf{W}$, target class $y$
**Output**: $\operatorname{argmax}_h [S_{\mathbf{W}}(\mathbf{x}, h) + \Delta(y, h)]$
**for** $r \in \{1, \ldots, R\}$ **do**
    $h^{(r)} := h^*(\mathbf{x}, \mathbf{W}, r, 1)$                      // using Alg. 2
    Let $\mathsf{Value}(r) := S_{\mathbf{W}}(\mathbf{x}, h^{(r)}), + \Lambda(y, r)$
Let $r^* = \operatorname{argmax}_r \mathsf{Value}(r)$
**return** $h^{(r^*)}$

---

### 5.1.3   Gradient update

Finally, we need to compute the gradient of the distance score. From (7), we have

$$\frac{\partial S_{\mathbf{W}}(\mathbf{x}, h)}{\partial \mathbf{W}} = \mathbf{\Psi}(\mathbf{x}, h) = -\sum_{\mathbf{x}_j \in h} (\mathbf{x} - \mathbf{x}_j)(\mathbf{x} - \mathbf{x}_j)^T. \quad (12)$$

Thus, the update in Alg 1 has a simple interpretation, illustrated in Fig 2 on the right. For every $\mathbf{x}_i \in h^* \setminus \widehat{h}$, it "pulls" $\mathbf{x}_i$ closer to $\mathbf{x}$. For every $\mathbf{x}_i \in \widehat{h} \setminus h^*$, it "pushes" it farther from $\mathbf{x}$; these push and pull refer to increase/decrease of Mahalanobis distance under the updated $\mathbf{W}$. Any other $\mathbf{x}_i$, including any $\mathbf{x}_i \in h^* \cap \widehat{h}$, has no influence on the update. This is a difference of our approach from

LMNN, MLR etc. This is illustrated in Figure 2. In particular $h^*$ corresponds to points $a$, $c$ and $e$, whereas $\widehat{h}$ corresponds to points $c$, $e$ and $f$. Thus point $a$ is pulled while point $f$ is pushed.

Since the update does not necessarily preserve $\mathbf{W}$ as a PSD matrix, we enforce it by projecting $\mathbf{W}$ onto the PSD cone, by zeroing negative eigenvalues. Note that since we update (or "downdate") $\mathbf{W}$ each time by matrix of rank at most $2k$, the eigendecomposition can be accomplished more efficiently than the naïve $O(d^3)$ approach, e.g., as in [17].

Using first order methods, and in particular gradient methods for optimization of non-convex functions, has been common across machine learning, for instance in training deep neural networks. Despite lack (to our knowledge) of satisfactory guarantees of convergence, these methods are often successful in practice; we will show in the next section that this is true here as well.

One might wonder if this method is valid for our objective that is not differentiable; we discuss this briefly before describing experiments. A given $\mathbf{x}$ imposes a Voronoi-type partition of the space of $\mathbf{W}$ into a finite number of cells; each cell is associated with a particular combination of $\widehat{h}(\mathbf{x})$ and $h^*(\mathbf{x})$ under the values of $\mathbf{W}$ in that cell. The score $S_\mathbf{W}$ is differentiable (actually linear) on the interior of the cell, but may be non-differentiable (though continuous) on the boundaries. Since the boundaries between a finite number of cells form a set of measure zero, we see that the score is differentiable almost everywhere.

## 6 Experiments

We compare the error of $k$NN classifiers using metrics learned with our approach to that with other learned metrics. For this evaluation we replicate the protocol in [11], using the seven data sets in Table 1. For all data sets, we report error of $k$NN classifier for a range of values of $k$; for each $k$, we test the metric learned for that $k$. Competition to our method includes Euclidean Distance, LMNN [22], NCA, [8], ITML [5], MLR [14] and GB-LMNN [11]. The latter learns non-linear metrics rather than Mahalanobis.

For each of the competing methods, we used the code provided by the authors. In each case we tuned the parameters of each method, including ours, in the same cross-validation protocol. We omit a few other methods that were consistently shown in literature to be dominated by the ones we compare to, such as $\chi^2$ distance, MLCC, M-LMNN. We also could not include $\chi^2$-LMNN since code for it is not available; however published results for $k = 3$ [11] indicate that our method would win against $\chi^2$-LMNN as well.

Isolet and USPS have a standard training/test partition, for the other five data sets, we report the mean and standard errors of 5-fold cross validation (results for all methods are on the same folds). We experimented with different methods for initializing our method (given the non-convex objective), including the euclidean distance, all zeros etc. and found the euclidean initialization to be always worse. We initialize each fold with either the diagonal matrix learned by ReliefF [12] (which gives a scaled euclidean distance) or all zeros depending on whether the scaled euclidean distance obtained using ReliefF was better than unscaled euclidean distance. In each experiment, $\mathbf{x}$ are scaled by mean and standard deviation of the training portion.[3] The value of $C$ is tuned on on a 75%/25% split of the training portion. Results using different scaling methods are attached in the appendix.

Our SGD algorithm stops when the running average of the surrogate loss over most recent epoch no longer descreases substantially, or after max. number of iterations. We use learning rate $\eta(t) = 1/t$.

The results show that our method dominates other competitors, including non-linear metric learning methods, and in some cases achieves results significantly better than those of the competition.

## 7 Conclusion

We propose a formulation of the metric learning for $k$NN classifier as a structured prediction problem, with discrete latent variables representing the selection of $k$ neighbors. We give efficient algorithms for exact inference in this model, including loss-augmented inference, and devise a stochastic gradient algorithm for learning. This approach allows us to learn a Mahalanobis metric with an objective which is a more direct proxy for the stated goal (improvement of classification by $k$NN rule)

| k = 3 | | | | | | | |
|---|---|---|---|---|---|---|---|
| Dataset | Isolet | USPS | letters | DSLR | Amazon | Webcam | Caltech |
| $d$ | 170 | 256 | 16 | 800 | 800 | 800 | 800 |
| $N$ | 7797 | 9298 | 20000 | 157 | 958 | 295 | 1123 |
| $C$ | 26 | 10 | 26 | 10 | 10 | 10 | 10 |
| Euclidean | 8.66 | 6.18 | 4.79 ±0.2 | 75.20 ±3.0 | 60.13 ±1.9 | 56.27 ±2.5 | 80.5 ±4.6 |
| LMNN | 4.43 | 5.48 | 3.26 ±0.1 | 24.17 ±4.5 | 26.72 ±2.1 | 15.59 ±2.2 | 46.93 ±3.9 |
| GB-LMNN | **4.13** | 5.48 | 2.92 ±0.1 | 21.65 ±4.8 | 26.72 ±2.1 | 13.56 ±1.9 | 46.11 ±3.9 |
| MLR | 6.61 | 8.27 | 14.25 ±5.8 | 36.93 ±2.6 | 24.01 ±1.8 | 23.05 ±2.8 | 46.76 ±3.4 |
| ITML | 7.89 | 5.78 | 4.97 ±0.2 | 19.07 ±4.9 | 33.83 ±3.3 | 13.22 ±4.6 | 48.78 ±4.5 |
| NCA | 6.16 | 5.23 | 4.71 ±2.2 | 31.90 ±4.9 | 30.27 ±1.3 | 16.27 ±1.5 | 46.66 ±1.8 |
| ours | 4.87 | **5.18** | **2.32 ±0.1** | **17.18 ±4.7** | **21.34 ±2.5** | **10.85 ±3.1** | **43.37 ±2.4** |

| k = 7 | | | | | | | |
|---|---|---|---|---|---|---|---|
| Dataset | Isolet | USPS | letters | DSLR | Amazon | Webcam | Caltech |
| Euclidean | 7.44 | 6.08 | 5.40 ±0.3 | 76.45 ±6.2 | 62.21 ±2.2 | 57.29 ±6.3 | 80.76 ±3.7 |
| LMNN | 3.78 | **4.9** | 3.58 ±0.2 | 25.44 ±4.3 | 29.23 ±2.0 | 14.58 ±2.2 | 46.75 ±2.9 |
| GB-LMNN | **3.54** | **4.9** | 2.66 ±0.1 | 25.44 ±4.3 | 29.12 ±2.1 | 12.45 ±4.6 | 46.17 ±2.8 |
| MLR | 5.64 | 8.27 | 19.92 ±6.4 | 33.73 ±5.5 | 23.17 ±2.1 | 18.98 ±2.9 | 46.85 ±4.1 |
| ITML | 7.57 | 5.68 | 5.37 ±0.5 | 22.32 ±2.5 | 31.42 ±1.9 | **10.85 ±3.1** | 51.74 ±2.8 |
| NCA | 6.09 | 5.83 | 5.28 ±2.5 | 36.94 ±2.6 | 29.22 ±2.7 | 22.03 ±6.5 | 45.50 ±3.0 |
| ours | 4.61 | **4.9** | **2.54 ±0.1** | **21.61 ±5.9** | **22.44 ±1.3** | 11.19 ±3.3 | **41.61 ±2.6** |

| k = 11 | | | | | | | |
|---|---|---|---|---|---|---|---|
| Dataset | Isolet | USPS | letters | DSLR | Amazon | Webcam | Caltech |
| Euclidean | 8.02 | 6.88 | 5.89 ±0.4 | 73.87 ±2.8 | 64.61 ±4.2 | 59.66 ±5.5 | 81.39 ±4.2 |
| LMNN | **3.72** | **4.78** | 4.09 ±0.1 | 23.64 ±3.4 | 30.12 ±2.9 | 13.90 ±2.2 | 49.06 ±2.3 |
| GB-LMNN | 3.98 | **4.78** | **2.86 ±0.2** | 23.64 ±3.4 | 30.07 ±3.0 | 13.90 ±1.0 | 49.15 ±2.8 |
| MLR | 5.71 | 11.11 | 15.54 ±6.8 | 36.25 ±13.1 | 24.32 ±3.8 | 17.97 ±4.1 | 44.97 ±2.6 |
| ITML | 7.77 | 6.63 | 6.52 ±0.8 | **22.28 ±3.1** | 30.48 ±1.4 | 11.86 ±5.6 | 50.76 ±1.9 |
| NCA | 5.90 | 5.73 | 6.04 ±2.8 | 40.06 ±6.0 | 30.69 ±2.9 | 26.44 ±6.3 | 46.48 ±4.0 |
| ours | 4.11 | 4.98 | 3.05 ±0.1 | **22.28 ±4.9** | **24.11 ±3.2** | **11.19 ±4.4** | **40.76 ±1.8** |

Table 1: $k$NN error, for $k$=3, 7 and 11. Features were scaled by z-scoring. Mean and standard deviation are shown for data sets on which 5-fold partition was used. Best performing methods are shown in bold. Note that the only non-linear metric learning method in the above is GB-LMNN.

than previously proposed similarity learning methods. Our learning algorithm is simple yet efficient, converging on all the data sets we have experimented upon in reasonable time as compared to the competing methods.

Our choice of Frobenius regularizer is motivated by desire to control model complexity without biasing towards a particular form of the matrix. We have experimented with alternative regularizers, both the trace norm of $\mathbf{W}$ and the shrinkage towards Euclidean distance, $\|\mathbf{W} - \mathbf{I}\|_F^2$, but found both to be inferior to $\|\mathbf{W}\|_F^2$. We suspect that often the optimal $\mathbf{W}$ corresponds to a highly anisotropic scaling of data dimensions, and thus bias towards $\mathbf{I}$ may be unhealthy.

The results in this paper are restricted to Mahalanobis metric, which is an appealing choice for a number of reasons. In particular, learning such metrics is equivalent to learning linear embedding of the data, allowing very efficient methods for metric search. Still, one can consider non-linear embeddings $\mathbf{x} \rightarrow \phi(\mathbf{x}; \mathbf{w})$ and define the distance $D$ in terms of the embeddings, for example, as $D(\mathbf{x}, \mathbf{x}_i) = \|\phi(\mathbf{x}) - \phi(\mathbf{x}_i)\|$ or as $-\phi(\mathbf{x})^T \phi(\mathbf{x}_i)$. Learning $S$ in the latter form can be seen as learning a kernel with discriminative objective of improving $k$NN performance. Such a model would be more expressive, but also more challenging to optimize. We are investigating this direction.

## Acknowledgments

This work was partly supported by NSF award IIS-1409837.

## Footnotes

[1]We discuss other choices of regularizer in Section 7.

[2]We note that both inference problems over $h$ are done in leave one out settings, i.e., we impose an additional constraint $i \notin h$ under the argmax, not listed in the algorithm explicitly.

[3]For Isolet we also reduce dimensionality to 172 by PCA computed on the training portion.

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
