[Supplementary Material]

# Discriminative Metric Learning by Neighborhood Gerrymandering (Supplementary Material)

**Shubhendu Trivedi,  David McAllester,  Gregory Shakhnarovich**
Toyota Technological Institute
Chicago, IL - 60637
{shubhendu,mcallester,greg}@ttic.edu

## 1  Proof of correctness of Algorithm 2

First of all it is easy to see that Algorithm 2 terminates. There are $k - n^*$ iterations after initialization (of the first $n^*$ points) and this amounts to at most a linear scan of $X$. We need $O(N \log N)$ time to sort the data and then finding $\mathbf{h}^*$ involves $O(N)$, thus the algorithm runs in time $O(N \log N)$.

We need to prove that the algorithm returns $h^*$ as defined earlier. First, we establish the correctness of setting $n^*$:

**Proposition 1.** *Let $R$ be the number of classes, and let $\#(h, y)$ be the count of neighbors from target class $y$ included in the assignment $h$. Then, $\Delta(y^*, h) = 0$ only if $\#(h, y^*) \geq n^*$, where*

$$n^* = \begin{cases} \left\lceil \frac{k+R-1}{R} \right\rceil & \text{if ties not allowed,} \\ \left\lceil \frac{k}{R} \right\rceil & \text{if ties allowed.} \end{cases}$$

*We prove it below for the case with no ties; the proof when ties are allowed is very similar.*

*Proof.* Suppose by contradiction that $\Delta(y^*, h) = 0$ and $\#(h, y^*) \leq \lceil \frac{k+R-1}{R} \rceil - 1$. Then, since no ties are allowed, for all $y \neq y^*$, we have $\#(h, y^*) \leq \lceil \frac{k+R-1}{R} \rceil - 2$, and

$$\sum_y \#(h, y) \leq (R-1) \left( \left\lceil \frac{k+R-1}{R} \right\rceil - 2 \right) \tag{1}$$

$$+ \left\lceil \frac{k+R-1}{R} \right\rceil - 1 \tag{2}$$

$$< k, \tag{3}$$

a contradiction to $|h| = k$. $\qquad\square$

Next, we prove that the algorithm terminates and produces a correct result. For the purposes of complexity analysis, we consider $R$ (but not $k$) to be constant, and number of examples from each class to be $O(N)$.

**Claim 1.** *Algorithm 2 terminates after at most $O((N + k) \log N)$ operations and produces an $h$ such that $|h| = k$.*

*Proof.* The elements of $X$ can be held in $R$ priority queues, keyed by $D_{\mathbf{W}}$ values, one queue per class. Construction of this data structure is an $O(N \log N)$ operation, carried out before the algorithm starts. To initialize $h$ with $n^*$ values, the algorithm retrieves $n^*$ top elements from the priority queue for class $y^*$. An $O(n^* \log N)$ operation. Then, for each of the iterations over $l$, the algorithm

needs to examine at most one top element from $R$ queues, which costs $O(\log N)$; each such iteration increases $|h|$ by one. Thus after $k - n^*$ iterations $|h| = k$; the total cost is thus $O(k \log N)$. Combined with the complexity of data structure construction mentioned above, this concludes the proof.

$\square$

Note that for typical scenarios in which $N \gg k$, the cost will be dominated by the $N \log N$ data structure setup.

**Claim 2.** *Let $h^*$ be returned by Algorithm 2. Then,*

$$h^* = \underset{h:|h|=k,\, \Delta(y^*,h)=0}{\mathrm{argmax}} S_{\mathbf{W}}(\mathbf{x}, h), \tag{4}$$

*i.e., the algorithm finds the highest scoring $h$ with total of $k$ neighbors among those $h$ that attain zero loss.*

*Proof.* From Proposition 1 we know that if $\#(h, y) < n^*$, then $h$ does not satisfy the $\Delta(\mathbf{x}, h) = 0$ condition. $|h| \geq n^*$ to (4) without altering the definition.

We will call $h$ "optimal for $l$" if

$$h = \underset{h:|h|=n^*+l,\, \#(h,y)\geq n^*,\, \Delta(y^*,h))=0}{\mathrm{argmax}} S_{\mathbf{W}}(\mathbf{x}, h).$$

We now prove by induction over $l$ that this property is maintained through the loop over $l$ in the algorithm.

Let $h^{(j)}$ denote choice of $h$ after $j$ iterations of the loop, i.e., $|h| = n^* + j$. Suppose that $h^{(l-1)}$ is optimal for $l - 1$. Now the algorithm selects $\mathbf{x}_a \in X$, such that

$$\mathbf{x}_a = \underset{\mathbf{x}_i:\, y_i=y,\, \text{or } \#(y_i)<\#(y)-\tau,\, \mathbf{x}_i \notin h}{\mathrm{argmin}} D_{\mathbf{W}}(\mathbf{x}, \mathbf{x}_i). \tag{5}$$

Suppose that $h^{(l)}$ is not optimal for $l$. Then there exists an $\mathbf{x}_b \in X$ for which $D_{\mathbf{W}}(\mathbf{x}, \mathbf{x}_b) < D_{\mathbf{W}}(\mathbf{x}, \mathbf{x}_a)$ such that picking $\mathbf{x}_b$ instead of $\mathbf{x}_a$ would produce $h$ optimal for $l$. But $\mathbf{x}_b$ is not picked by the algorithm; this can only happen if conditions on the $\mathrm{argmin}$ in (5) are violated, namely, if $\#(y_b) = \#(y) - \tau$; therefore picking $\mathbf{x}_b$ would violate conditions of optimality of $h^{(l)}$, and we get a contradiction.

It is also clear that after initialization with $k$ highest scoring neighbors in $y^*$, $h$ is optimal for $l = 0$, which forms the base of induction. We conclude that $h^{(k-n^*)}$, i.e. the result of the algorithm, is optimal for $k - n^*$, which is equivalent to definition in (4).

$\square$

## 2   Runtimes using different methods

Here we include the training times in seconds for one fold of each dataset. These timings are for a single partition, for optimal parameters for $k = 7$. These experiments were run on a 12-core Intel Xeon E5-2630 v2 @ 2.60GHz.

| Dataset | DSLR | Caltech | Amazon | Webcam | Letters | USPS | Isolet |
|---------|------|---------|--------|--------|---------|------|--------|
| LMNN | 358.11 | 1812.1 | 1545.1 | 518.7 | 179.77 | 782.66 | 1762.1 |
| GB-LMNN | 410.13 | 1976.4 | 1680.9 | 591.29 | 272.87 | 3672.9 | 2882.6 |
| MLR | 4.93 | 124.42 | 88.96 | 85.02 | 838.13 | 1281 | 33.20 |
| MLNG | 413.36 | 1027.6 | 2157.2 | 578.74 | 6657.3 | 3891.7 | 3668.9 |

# 3 Experimental results using different feature normalizations

**k = 3**