[Reviews · NeurIPS 2014]

Submitted by Assigned_Reviewer_35

The paper proposes a new approach for Mahalanobis metric learning for k-nearest neighbor (kNN) classification. The main difference from the existing work is in the way how k "nearest neighbors" are found. Instead of simply looking for the k nearest neighbors, the authors are searching for the closest k examples that also guarantee correct classification (the authors refer to it as the "gerrymandering"). They propose a greedy algorithm to find such a neighborhood. The class that results in the most compact such neighborhood is selected as a prediction. A gradient descent algorithm is used to learn a positive definite weight matrix that maximizes the proposed loss function. The experimental results show that the proposed algorithm is superior to several competing KNN algorithms.

Positives:
- the "gerrymandering" idea is interesting and seems to be novel
- the experimental results are promising

Negatives:
- the paper is not as easy to read as it could be, there are some typos in formulas (e.g., intersection is used instead of union, and such), the presentation is muddled and poorly organized (e.g., Figure 1 is explained too late and it does not provide too much of an insight as to why the approach works), appendix is referred to but not submitted
- the idea is heuristically-based and not theoretically characterized
Summary: The paper proposes an interesting "gerrymandering" idea for kNN-like classification and a gradient descent method for Mahalanobis metric learning. On the negative side, the presentation is a weak point of the paper and the approach is not theoretically characterized.

Submitted by Assigned_Reviewer_46

This paper presents a structured learning approach
to metric learning for nearest neighbour classification. The
authors develop the corresponding structured formulation and
present algorithms to solve for the (loss augmented) inference
problems. On some common datasets they outperform reasonable
baselines.

The main idea presented in this paper is to pose metric learning
as a problem where the objective should be to get good nearest
neighbour classification results. This translates into not all
points of the same class need to be close together, but for
every point just a set of \ceil{k} neighbours. The problem for
this approach becomes the inference problem, that is to search
for the offending set of nearest neighbours that flip the
prediction and are close to the point in question. The authors
present algorithms for both the inference and loss-augmented
inference problem.

This is a simple and appealing idea that, evidenced in the
experimental section, does work. The paper is well written, a
simple approach well presented and easy to follow. Some more
details could be given on the experimental part, runtime,
dependency on local minima, etc. Also, the paper will benefit
from another proof reading.

Summary: A simple idea to learn metrics for nearest neighbour
prediction. Technical contributions for a sensible formulation
are presented and some experiments demonstrate proof-of-concept
and better performance compared to reasonable baselines.

Submitted by Assigned_Reviewer_47

The paper proposes a new approach to Mahalanobis metric learning which views metric learning as a structure-prediction problem. The structure we seek to establish is the set, $h$, of appropriate k-Nearest Neighbors (k-NNs) of an instance; appropriate in the sense that it will lead to a small classification error. The idea of the authors is quite elegant: in learning a metric for k-NN prediction we can have a correct prediction for an instance $x_i$ as long as the majority of its k-NNs are of the same class as $x_i$. The metric learning algorithm they propose does exactly that, i.e. it looks for the Mahalanobis metric matrix $W$ that leads to k-nearest neighborhoods for each $x_i$ instance such that the majority of the instances of the neighborhood belongs to the correct class, i.e. that of $x_i$. Note here that there can be many such neighborhoods that predict the correct label for a given instance $x_i$, it is enough to prefer at least one of them over all the ones that predict the wrong label.

The authors define a structure prediction problem in which they optimize a tractable surrogate loss which is parametrized as a Mahalanobis distance in which for each instance the structure sought is the set of instances $h$ k-NNs that will have zero classification error. The authors build directly on the work of [12], McFee and Lankriet "Metric Learning to rank", ICML 2010, where metrics were learned to predict another type of structure, related to the k-NN, rankings. They use the same feature map $\Psi(x,h)$ to quantify the relation of an instance $x$ to the $h$ structure (which is defined as the matrix that contains the negatives of the sums of the squared pairwise attribute differences of $x$ and its neighbors in $h$). This map under the Frobenious inner product with a PSD matrix $W$ leads to the sum of the negative Mahalanobis distances of $x$ from the instances of $h$, measuring in a sense the similarity of $x$ to $h$, S_W(x,h). The set of instances that maximizes S_W is the set of k-NNs of $x$, and the standard classification loss is the difference between the true label $x$ and the majority label of $h. Since this is intractable the authors define a tractable loss on the basis of the neighborhood that mostly violates the correct classication, using the sum of its loss (which plays the role of the margin) and its similarity to $x$, and the neighborhood that results in a correct classification and has the largest similarity to $x$.

The minimize the loss together with a frobenius norm regularizer on $W$. The minimization is done by stochastic gradient descent at every step of which they solve two neighborhood inference problems. One inference problem finds the neighborhood that has maximum similarity and classiciation-loss (loss augmented inference) and the other finds the maximum similarity zero classification-loss neighborhood. The gradient update of the loss is then computed over these two inferenced values. The update rule learns a W such that it pulls instances close together if they belong in the zero classification loss neighborhood or far apart if the belong in a neighborhood that mostly violates the correct classication constraints, all other instances do not influence the updates. At each step there is a projection on the PSD matrix cone.

In the experimental section the authors compare their method against LMNN, MLR (the metric learning to rank method), and the gradient boosted variant of LMNN on six datasets with small to medium numbers of instances, up to 20k and a few hundreds of features. The evaluated the performance of the different methods under k-NN, with k-NN=3,7,11. The empirical results show an advantage of their method over the baselines.

Overall I quite liked the idea presented in the paper and the definition of metric learning in which one has to find the right neighborhood, formulating the problem as a structure prediction problem. Even though it builds considerably on [12] I think that the optimization over the neighborhoods makes it substantially different from that paper, as well as interesting. One of the critisisms of the prior work and in particular LMNN is the fact that the target neighborhood structure remains fixed over the optimization. There are extensions of LMNN that do the update of the neighborhood as a part of learning, lifting that constraint, e.g. Jun Wang et al, Learning Neighborhoods for Metric Learning, ECML/PKDD 2012; the authors should discuss them and ideally include them as a baseline in the experiments since they also learn the neighborhood.

I am missing a discussion on the computational complexity especically in view of the small to medium size of the datasets used in the evaluation. How fast does the method converge, what are the problem sizes to which it scales. Also the experimental section is a bit weak, the hyper-parameters of the different methods as well as the grids from which they take values are not described fully. In addition I would have welcomed a more extensive methods with other metric methods which have been shown to have good predictive performance, for example Jun Wang et al, Parametric Local Metric Learning for Nearest Neighbor Classification, in NIPS 2012, which has been shown to have better performance than the baseline methods against which the authors compare.
Summary: A new metric learning problem that formulates the metric learning as structure learning problem where the structure that is learned is the set of nearest neighbors. The experimental section could be strengthened.

Submitted by Assigned_Reviewer_48

This paper proposes a novel metric learning model leveraging the objective of KNN. As most other metric learning methods, this paper uses Mahalanobis metrics. Its parameters are optimized with stochastic gradient descent. For a given point, its loss function corresponds to the difference between the "worst" k neighbors and the "best" k neighbors, which is quite novel and meaningful. Unlike LMNN trying to push all bad neighbors (imposters) away and pull all good neighbors (targets) in where imposters and targets are selected before learning, minimizing this loss is equivalently to only pushing some bad neighbors away from the given point and pulling some good neighbors in so that the resulting KNN prediction would become correct, as illustrated in Figure 1. However, the use of KNN objective comes at a price, because the "worst" offending neighbors and the "best" neighbors should be chosen dynamically. The authors propose Algorithm 2 and 3 to try to address this problem with complexity O(NlogN), which is still a great cost from my perspective since these two algorithms should be used in every gradient iteration. The experimental results demonstrate the good performance of the proposed method. But the datasets are all of small or middle scale. Could you show or give a sense of how fast the training was? Also, it seems a bit counter-intuitive that a bigger k results in less advantage of the proposed method over the other methods, since I guess a bigger k gives more freedom to the proposed method to manipulate the neighbors than the other methods.

A very big downside of this paper is that no appendix is attached, resulting in many question marks in the section 5.1.1. This is not a minor problem as it also makes reviewers hard to verify the correctness of Algorithm 2 and 3.

Minor problems:
1. There's a big white margin at the beginning of page 3 (between the first paragraph and the title of section 2)
2. To me, Figure 1 shows up earlier than it's supposed to be. Moving figure 1 further behind may also fix the huge margin as mentioned above.
3. Line 67, "x has radius equal to th e farthest of ...", "th e" --> "the"
4. Line 314, there is an extra comma

After reading the author feedback, I think efficiency is still an issue for the proposed algorithm. From the reported running time, I can see an approximately linear increase of running time as the number of training instances increases. But it's unclear whether this linearity would preserve on larger data sets. A parallel version may be desired.
Summary: The ideas proposed in this paper are great and novel though I think the training efficiency might be a problem. Please also submit the appendix so that we can also verify the correctness of the algorithms.
Author Feedback
Author rebuttal: We appreciate the detailed reviews. Three issues brought up by multiple reviewers:

1) Appendix. We initially included it but due to our technical mistake it was removed. At our request the chairs have kindly restored it, and the supp. materials in the CMT now contains full version, including the appendix which has proof of correctness for Algorithm 2. We are sorry for this glitch.

2) Timing information. We will include it in the paper. Briefly, our method trains in about the same time as GB-LMNN, and is a bit slower than LMNN.
The training times (for k=7, optimal value of C):DSLR: 7min
Caltech: 17min
Amazon: 36min
Webcam: 10min
Letters: 111min
USPS: 65min
Isolet: 61min

The fastest method of the ones we experiment with is MLR. We are looking into borrowing some of the optimization techniques used in MLR code to speed up our implementation (which is currently written somewhat naively, entirely in Matlab).

We have also compared our method, since the original submission, to ITML and NCA, and both of those are produce worse accuracies than our method. NCA is significantly slower than our method, while ITML is a lot slower for data of high dimensionality (but faster on low-dim data sets).

3) Typos. We apologize for these, and will of course fix them as well as follow suggestions by reviewers to improve exposition (e.g., move Figure 1).

Response to other comments:
R47: We will include the work by Jun Wang et al. in discussion, and work in including it in experimental comparison.

R48: Yes, generally higher k could lead to more flexibility in gerrymandering and thus potentially higher gains. That is limited however, by properties of a particular data set: the number of examples from the correct class, and by the geometry of the position of the examples in the space, which is changed by the learned Mahalanobis metric only up to a linear transformation. And if there are only a few examples from the target class in the data set (which is the case in some of the data sets we experiment on), then often only a few of them can be brought into a Mahalanobis neighborhood without “dragging in” too many examples from a wrong classes. This leads to a “saturation” at a low value of k, increasing much past which would not produce any gains.